# Modeling water balance components of conifer species using the Noah-MP model in an eastern Mediterranean ecosystem

Mohsen Amini Fasakhodi[a], Hakan Djuma[a], Ioannis Sofokleous[a], Marinos Eliades[a,1], Adriana Bruggeman[a]

[a] Energy, Environment and Water Research Center, The Cyprus Institute, 20 Konstantinou Kavafi Street, 2121 Aglantzia,

Nicosia, Cyprus

[1] Present address: Eratosthenes Center of Excellence, 82 Franklin Roosevelt, 3012 Lemesos, Cyprus

Correspondence to:  Mohsen Amini Fasakhodi (m.amini-fasakhondi@cyi.ac.cy)

## Abstract

Few studies have investigated the performance of land surface models for semi-arid Mediterranean forests. This study aims

to parameterize and test the performance of the Noah-MP land surface model for an eastern Mediterranean ecosystem. To

this end, we modeled the water balance components of two conifer species, *Pinus brutia* and *Cupressus sempervirens,* in a

plantation forest on the Mediterranean island of Cyprus. The study area has a long-term average annual rainfall of 315 mm.

Observations from 4 sap flow and 48 soil moisture sensors, for the period from December 2020 to June 2022, were used for

model parameterization. A local sensitivity analysis found that the surface infiltration (*REFKDT*), hydraulic conductivity

(*SATDK*) and stomatal resistance (*RSMIN*) parameters had the highest impacts on the soil water balance components and

transpiration. The model performed better during the wetter 9-month validation period (379 mm rain) than during the drier

10-month calibration period (175 mm rain). Average soil moisture in the top 60-cm of the soil profile was reasonably well

captured for both species (daily *NSE* > 0.70 for validation). Among the three soil layers, the second layer (20-40 cm) showed

better simulation performance during both periods and for both species. The model exhibited limitations in simulating

transpiration, particularly during the drier calibration period. Inclusion of a root distribution function in the model, along

with the monitoring of soil moisture below the 60-cm soil depth in the field, could improve the accuracy of model

simulations in such water-limited ecosystems.

**Keywords:** semi-arid region, sap flow, soil moisture, Land surface model, Noah-MP, WRF

## 1. Introduction

Evapotranspiration, which is a combination of evaporation from soil and vegetation and transpiration from vegetation, plays

a crucial role in the terrestrial water cycle (Zhan et al., 2019) and is recognized as the most determinative component of the

water balance in Mediterranean ecosystems (Corona and Montaldo, 2020). Evapotranspiration is controlled by atmospheric

demand and water availability, which are in turn influenced by climate, vegetation, and soil conditions (Wang et al., 2015).

Evapotranspiration is a key component of the hydrological cycle and the land surface energy balance and, globally, returns

back around 67% of the precipitation to the atmosphere (Zhang et al., 2016). In Mediterranean pine forests, the fraction of

precipitation that becomes evapotranspiration could amount to 90% or even 100% when trees extract water from deeper soil

layers (Ungar et al., 2013). Hence, knowledge of forest water balances in semi-arid regions and a better comprehension of its

drivers are important for improving the performance of land surface models (Lu et al., 2022) and for developing sustainable

land and water management policies in these regions (Vicente et al., 2018).

Pine forests form an important component of the Mediterranean landscape (Ganopoulos et al., 2013). *Pinus brutia* and *Pinus*

*halepensis* forests cover more than 7 million ha around the Mediterranean basin and perform important ecological and

economic roles in low- to mid-elevation forests (Chambel et al., 2013). *P. halepensis* mainly occupies the most southern and

western parts of the Mediterranean Basin, while *P. brutia* is confined to the eastern Mediterranean (Chambel et al., 2013).

However, these two species are very similar and were historically considered as two varieties of *P. halepensis,* although later

they were recognized as distinct species based on morphological and biochemical differences (Boydak 2004). In addition to

the great ecological value of natural stands, these Mediterranean pines may provide a highly resilient forest cover under dry

conditions for better rainfall infiltration, soil stabilization, and timber production (Chambel et al., 2013). *Cupressus*

*sempervirens (Mediterranean cypress)* is another important native coniferous species in the eastern Mediterranean (Bagnoli

et al., 2009). The species has a wide distribution across the Mediterranean region and plays a significant role in the local

ecosystem, economy, symbolism, and culture (Bagnoli et al., 2009). The beneficial ecosystem services provided by these

species, such as mitigating the urban heat island effect, carbon sequestration, soil conservation, drought toleration, and

habitat provision for wildlife, have been reported in many papers (e.g., Boydak *2004*; Kostopoulou et al., 2010, Helman et al., 2017a).

Despite numerous water balance-related studies in Mediterranean forest areas (e.g., Molina et al., 2012; Montaldo et al., 2021; Simpson et al., 2022), only a few studies have focused on quantifying water balance components of *C. sempervirens, P. brutia* and *P. halepensis* species (e.g., Yaseef et al., 2010; Ungar et al., 2013; del Campo et al., 2014; Helman et al., 2017b; Eliades et al., 2018; Rohatyn et al., 2018). Helman et al. (2017b) estimated annual evapotranspiration in a planted pine forest (*P. halepensis*) in Israel (279 mm average annual rainfall), using eddy covariance observations and a remote sensing-based model (RS–Met). Eliades et al. (2018) used sap flow measurements from eight trees, soil moisture and throughfall observations and a water balance model to calculate the daily water balance components of a stand of homogenous pine forest (*P. brutia*) in the Troodos mountains of Cyprus under 425 mm average annual rainfall. Rohatyn et al. (2018) used a mobile lab (with flux measurements and meteorological sensors) and a permanent flux-tower to compute fluxes of water, energy, and carbon in *P. halepensis* forests in Israel. They used multiple stepwise regression models to extend campaign data collected during 4-years to annual and interannual time series. However, no research in the literature has combined field observations with land surface models (LSMs) to estimate water balance components in Mediterranean coniferous forests.

Land surface models have been extensively used to evaluate and predict water fluxes because of their mechanistic-based structure and their application over a wide range of spatiotemporal scales (Chen et al., 2013). Through decades of development, LSMs have become more comprehensive and the current, third-generation models represent an increasing number of interactions and feedbacks between physical, biological, and chemical processes (Niu et al., 2011). Noah-MP is an augmented version of the Noah LSM (EK et al., 2003) and belongs to a new generation of LSMs that utilize a multi-parameterization framework that allows the use of different combinations of physical schemes for the different physical processes on the land surface (Chen et al., 2016). The Noah-MP model is the default land surface model within the widely used Weather Research and Forecasting (WRF; Skamarock et al., 2019) model, simulating all land-atmosphere interactions.

The Noah-MP model performance to estimate water balance components was assessed against a number of other LSMs with

satisfactory results (e.g., Cai et al., 2014; Chen et al., 2014; Sun et al., 2021).   Despite advances in LSMs, there are still

uncertainties in the simulation and partitioning of evapotranspiration. This may be rooted in uncertainties in the atmospheric

forcing and land surface properties data sets, parameterizations of the physical and biogeochemical processes, and the

spatiotemporal scales and resolutions of the simulations (Liu et al., 2015). To decrease the uncertainties in LSM applications,

an informative initial step is to identify sensitive model parameters for specific model outputs and then to calibrate these

parameters using local observations (Cuntz et al., 2015). Meir and Woodward (2010) also noted that results obtained from

LSMs without evaluation with in-situ measurements are questionable.

In general, there is a lack of information on the water balance components of coniferous species in Mediterranean

ecosystems. The performance of Noah-MP, which can be used to model these water balance components, has never been

specifically evaluated for a semi-arid Mediterranean ecosystem. To this end, the overall goal of this study is to model the

water balance components of two typical coniferous species, *P. brutia* and *C. sempervirens*, within an eastern Mediterranean

ecosystem, using the Noah-MP land surface model. The study has three specific objectives: (1) to evaluate the impact of

changes in model input parameters on tree water balance components (tree transpiration, soil evaporation and total runoff);

(2) to calibrate the most impactful parameters and evaluate the model's performance, using observed and modeled tree

transpiration and soil moisture values; and (3) to assess differences in the water balance components simulated with  the

calibrated Noah-MP model and with the Noah-MP default settings of the WRF model. The study utilizes a one-dimensional,

single grid-cell version of the Noah-MP model, applied to the rootzone area of a single tree.

## 2. Data and methods

### 2.1. Site description and measurements

Observations from a tree plantation in Athalassa Forest Park on the south-eastern edge of the city of Nicosia in Cyprus

(35.133° N, 33.400° E) were used for this study. The experimental site is located on a sedimentary formation with a sandy

loam soil texture. Percussion drilling, at three random locations, showed soil depths of approximately 1 m. The field is

relatively flat (with a mean slope of 4%) and is covered by a combination of seasonal vegetation and indigenous trees and



shrubs with a 5 to 6-m planting distance, which covers a 10-ha area. The long-term (1980–2010) mean annual rainfall

recorded at the nearby Athalassa meteorological station is 315 mm. Figure 1 presents the study area and the two studied

species.

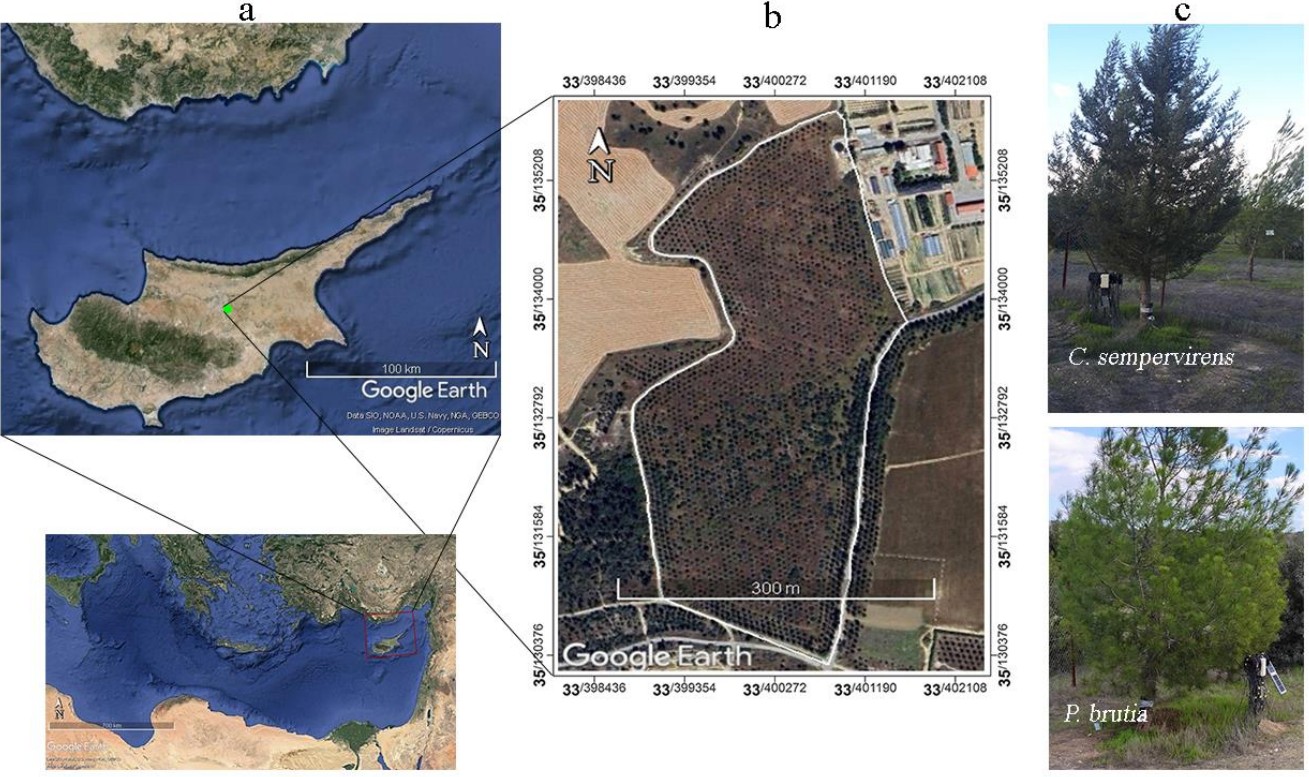

**Figure 1: The eastern Mediterranean with the island of Cyprus (red framed) and the study area location identified
with a green bullet (a); the study area in Athalassa National Forest Park framed with a white polygon (b); and two of
the monitored tree species with sap flow and soil moisture sensors (c). © Google Earth.**

Meteorological data were collected at the site in 5-minute intervals with an all-in-one weather sensor (ATMOS 41, METER

Group, Muenchen, Germany). To run the model during the calibration and validation periods, we used instantaneous

meteorological forcing data at 30-minute intervals, with the exception of precipitation, which was summed up. These

variables include air temperature (K), relative humidity (kg kg$^{-1}$), downward solar radiation (W m$^{-2}$), downward longwave

radiation (Wm$^{-2}$), precipitation (Kg m$^{-2}$s$^{-1}$), wind speed (m s$^{-1}$), and atmospheric pressure (hPa). We used data from two other

meteorological stations to fill gaps in our ATMOS 41 data time series: (i) the Athalassa governmental meteorological station,

located around 1 km from our experimental site; and (ii) the meteorological sensors of an eddy covariance tower

instrumented at a height of 2 m in a neighboring agricultural field. For the model sensitivity analysis, 30-minute

meteorological forcing data were extracted from optimized WRF simulations for the 1-km grid cell covering the site, for

October 2016 to September 2019 (Sofokleous et al. 2021).

We used observations of transpiration and soil moisture from two *P. brutia* and two *C. sempervirens* trees between

December 2020 to June 2022 for the calibration and validation of the Noah-MP model. We will refer to these species as pine

and cypress as the representative species of the field. Sap flow sensors (SFM1, ICT International, Armidale, Australia) were

placed on the trees at a height of approximately 30 cm and used to measure the sap flow velocity at hourly intervals. The

sensor observations were processed to calculate tree transpiration with the Sap flow tool software, following the methods of

Burgess et al. (2001). Soil moisture (volumetric soil water content) and soil temperature were measured hourly with soil

moisture sensors (SMT100, Truebner, Neustadt, Germany). These sensors were installed at depths of 10, 30, and 50 cm and

were placed around each tree in two opposite directions. In each direction, one set of sensors was installed under the tree

canopy (approximately 0.7 m from the tree trunk) and the other set at the edge of the canopy (approximately 1.8 m from the

trunk). Thus, for each tree species, the average daily sap flow of the two trees and the average soil moisture from eight

sensors at each depth were used for the model calibration and validation.

Average soil porosity at the site was 0.43 $cm^3/cm^3$, calculated from bulk density samples collected from six locations and

two depths (0-10 and 40-50 cm). Field capacity and wilting point were derived from the soil moisture observations. The field

capacity (0.18 $cm^3/cm^3$) was taken from a very wet period in early February 2022, after drainage had stopped, and the

wilting point (0.06 $cm^3/cm^3$) was the lowest observed soil moisture level during the summer months.

### 2.2. The Noah-MP Land Surface Model

The Noah-MP model (Niu et al., 2011; Yang et al., 2011) has been developed through the integration of different physics

modules from other land surface models (Kumar et al., 2017). Many options considered in the model represent interaction

processes between the land surface and the atmosphere (Barlage et al., 2015). A variety of formulations are used in these

options to calculate processes such as soil parameterization, runoff generation, stomatal conductance, and radiative transfer

in the canopy (Cuntz et al., 2016).

### 2.2.1. Model description and setup

In the model version used in this study (version 1.1), there are 12 different physics schemes, each with different options that users can set based on their objectives and environmental conditions. We selected the same options as used by Sofokleous et al. (2023) in their WRF-Hydro (version 5.1) application for Cyprus (see Table A1). These authors selected the Jarvis option for the canopy stomatal resistance scheme, because the alternative Ball-Berry option was found to underestimate transpiration. Options for other schemes were selected consistent with WRF-Hydro's recommended default options (Gochis et al., 2020). We entered a constant leaf area index (*LAI*) for each species for the monthly *LAI* values in the input tables, because *LAI* observations of the pine and cypress trees at the site showed little seasonal variation.

The default soil column in the Noah-MP model is a 2-m soil column, which is discretized into four layers (10, 30, 60, and 100 cm from the top to the bottom of the soil column). Based on field observations, we changed the soil depth to 1 m and set the thickness of the layers to 20, 20, 20, and 40 cm. We assumed that all four layers had tree roots.

The model was Initialized by soil moisture and soil temperature recorded in the field. The initial conditions for the sensitivity analysis were taken from October 2020. The initial volumetric soil moisture (*SMC*) of the four soil layers was, from top to bottom, 0.06, 0.07, 0.07, and 0.07; and the soil temperature was, 25.7, 25.9, 26.2, and 26.4 °C. The skin temperature was set as the average of the upper layer soil temperature and the air temperature (21.4 °C).

### 2.2.2. Model conceptualization

We modeled a single tree with its rootzone as a single grid cell. The grid cell represents the rootzone area from which the tree extracts water. If all trees at the site are equal, the maximum rootzone extent of a tree (grid cell area) would be equal to the planting area of the trees (30 m³). Considering the relatively small canopy areas of the trees (6.2-7.3 m²), it is likely that the roots are not extending into the full planting area. Because the actual root extension area is unknown, we tested different grid cell areas (see Sect. 2.2.3), using the below equations. The vegetation fraction of the grid cell (*FVEG* (-)) was determined as follows:



$$FVEG = CA/GA \tag{1}$$

where $CA$ is the observed canopy cover area (m$^2$), and $GA$ is the grid cell area (m$^2$). The leaf area index ($LAI$ (m$^2$/m$^2$)) of the grid cell was determined as follows:

$$LAI = LAI_{CA} \times FVEG \tag{2}$$

where $LAI_{CA}$ is the $LAI$ of the canopy area. The planting area was 30 m$^2$ (5 m x 6 m). The average observed $CA$ was 7.3 m$^2$ for the two pine trees and 6.2 m$^2$ for the two cypress trees. The observed $LAI_{CA}$ was 4.4 for pine and 4.6 for cypress. Figure 2 presents a schematic of the modeled tree, rootzone, and the water balance components.

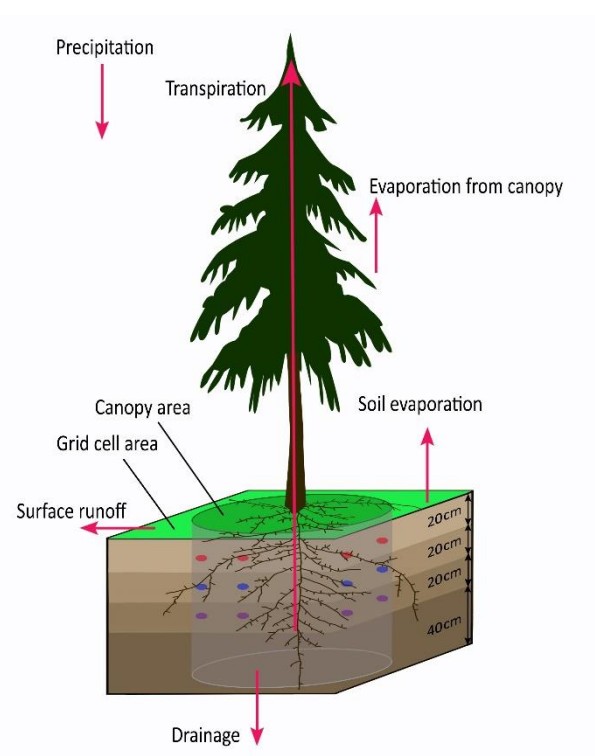

**Figure 2: Schematic of a modeled tree with canopy cover area, grid cell area, the soil column with its discretized layers, and the water balance components (red arrows); Red, blue and purple bullets in the tree root zone show the location of the soil moisture sensors at 10-, 30-, and 50-cm soil depth respectively. Horizontal and vertical dimensions are not scaled.**

### 2.2.3. Sensitivity analysis

Noah-MP calculates the different flux exchanges between the biosphere and the atmosphere for different plant functional types and different soil textures. There are 49 parameters for each plant functional type in the model and 10 for each soil





texture. However, not all parameters are used by all physics schemes. For our selected options there are 18 parameters involved in the modelling of one plant functional type with one soil texture (see Table 1). We analyzed the impact of the selected parameters on three output variables, namely tree transpiration, soil evaporation, and total runoff, using a local

sensitivity analysis. In this method, each parameter is changed in turn, while keeping all other parameters constant at their baseline values.

**Table 1: Vegetation and soil input parameter baseline, minimum and maximum values used for the sensitivity analysis; and their range as specified in the model input parameter files for evergreen needle-leaf forest and for soil textures ranging from sand to loam.**

| Parameter | Definition (unit) | Range | Baseline | Min | Max |
|---|---|---|---|---|---|
| **Vegetation parameters** | | | | | |
| FVEG* | Vegetation fraction (m²/m²) | 0-1 | 0.5 | 0.25 | 1 |
| LAIM* | Monthly maximum leaf area index (m²/m²) | 0-5.5 | 1.25 | 0.625 | 2.5 |
| CH2OP | Maximum intercepted water per leaf and stem area (mm) | 0.05-0.5 | 0.1 | 0.05 | 0.5 |
| HVT | Height of top of canopy (m) | 1-20 | 4 | 2 | 8 |
| RSMIN | Minimum stomatal resistance (s/m) | 40-300 | 171 | 125 | 410 |
| RSMAX | Maximum stomatal resistance (s/m) | - | 5000 | 2000 | 6000 |
| RGL | Radiation stress parameter (-) | 30-100 | 30 | 30 | 100 |
| HS | Vapor pressure deficit parameter (-) | 36.25-55 | 47.35 | 36.35 | 55 |
| TOPT | Optimum transpiration air temperature (K) | - | 298 | 292 | 298 |
| NROOT | Number of soil layers with roots (-) | 0-4 | 4 | 3 | 4 |
| **Soil parameters** | | | | | |
| SATDK | Saturated soil hydraulic conductivity (mm/h) | 3.6-507.6 | 18.8 | 3.9 | 50.8 |
| MAXSMC | Porosity (m²/m²) | 0.20-0.48 | 0.43 | 0.34 | 0.48 |
| SATPSI | Saturated soil matric potential (m) | 0.04-0.76 | 0.14 | 0.04 | 0.76 |
| BEXP | Pore size distribution index (-) | 2.79-11.55 | 4.74 | 2.8 | 5.3 |
| REFSMC | Field capacity (m²/m²) | 0.17-0.45 | 0.18 | 0.24 | 0.38 |
| WLTSMC | Wilting point soil moisture (m²/m²) | 0.01-0.138 | 0.06 | 0.03 | 0.08 |
| REFKDT | Surface infiltration parameter | - | 3 | 0.5 | 5 |
| SLOPE | Drainage parameter | 0-1.0 | 0.1 | 0.01 | 0.2 |

*FVEG, and LAIM were set in relation to each other with Equations 1 and 2.





We used a Relative Change ($RC$) equation to calculate the impact of the change in a selected input parameter values on the output variables:

$$RC = (O - O_b)/O_b \qquad (3)$$

where $RC$ is the relative change, and $O_b$ and $O$ denote the model output value at the baseline value of the input parameter and at a specific value of the input parameter, respectively. To rank each input parameter based on their impacts on the three outputs, we averaged their $RCs$ for all outputs and all three years. However, we chose to focus on parameters that had an impact on at least one output or in at least one year, in order to ensure that our analysis was relevant and meaningful.

In Table 1, the range column shows the ranges of parameter values as specified for evergreen needle-leaf forests and for
sandy and loamy soil textures in the model input table files. For the sensitivity analysis, we use baseline, minimum and maximum values, which represent the best estimates for our study area, based on field observations and data from the literature. To analyse the effect of the root extension area, we examined three different cases in which $FVEG$ and $LAI$ were defined using Eq. (1) and Eq. (2), using the average canopy cover area (7.3 m$^2$) and $LAI_{CA}$ (4.4) of the pine trees. For the maximum run, we set $FVEG$ equal to 1, which represents the case where the grid cell area equals the canopy cover area (7.3
m$^2$). For the minimum run, we set $FVEG$ equal to 0.25, representing a grid cell area close to the planting area (29.2 m$^2$). For the baseline run $FVEG$ was set equal to 0.5. We used the two-stream radiation option as our baseline setting and tested relative changes due to the use of the modified two-stream option of the radiation transfer scheme in one run. Finally, we examined the impact of the first soil layer thickness on the output variables by changing it around its baseline value of 20 cm (min = 10 cm, max = 30 cm).

The baseline and maximum values for minimum stomatal resistance ($RSMIN$) and minimum and maximum values for optimum transpiration air temperature ($TOPT$) were based on flux observations and model simulations of a *Pinus pinea* forest near Pisa in Italy (Hoshika et al., 2017). Minimum and maximum values for radiation stress ($RGL$) and vapor pressure deficit ($HS$) parameters were obtained from observations and Noah model simulations by Hogue et al. (2006) in Arizona. Maximum and minimum values for maximum intercepted water per leaf and stem area ($CH2OP$) and the surface infiltration
factor ($REFKDT$) were based on Niu et al. (2011).



### 2.2.4. Parameter calibration and model validation

A stepwise trial and error method was applied to calibrate all parameters with an absolute *RC* of 0.05 (5%) or higher for each of the three water balance outputs, occurring in any of the three years. Initially, we examined combinations of the minimum and maximum values of the selected parameters, as presented in Table 1. We then refined these values in the following steps.

We tested the fit of modeled and observed daily total transpiration and instantaneous soil moisture of the top 60-cm soil at 0:00 hour (average of three layers), using four evaluation criteria, namely, *BIAS*, mean absolute error (*MAE*), Nash-Sutcliffe efficiency (*NSE*) and Kling-Gupta efficiency (*KGE*). The *BIAS* for transpiration is computed as the sum of the daily errors. For soil moisture, considering that the daily values are not independent of each other, *BIAS* is calculated as the average daily error. The formulations for these criteria are as follows

$$BIAS = \sum_{i=1}^{n}(X_i - Y_i) \text{ for transpiration} \tag{4}$$

$$BIAS = \sum_{i=1}^{n}(X_i - Y_i)/n \text{ for soil moisture} \tag{5}$$

$$MAE = \sum_{i=1}^{n}|X_i - Y_i|/n \tag{6}$$

$$NSE = 1 - \left(\sum_{i=1}^{n}(Y_i - X_i)^2 / \sum_{i=1}^{n}(X_i - \bar{X})^2\right) \tag{7}$$

$$KGE = 1 - \sqrt{(r-1)^2 + ((\beta - 1)^2 + (\gamma - 1)^2} \tag{8}$$

where, $n$ is the number of days, $X_i$ and $Y_i$ denote observed and modeled values of the $ith$ day, $\bar{X}$ is the mean of observed values, $r$ is the Pearson correlation coefficient between observed and simulated values, $\beta$ is the ratio of the standard deviation of the simulated values to the standard deviation of the observed values, and $\gamma$ is the ratio of the mean of the simulated values to the mean of the observed values. Given the last three components, linear correlation, temporal variability, and mean bias contribute to the $KGE$ metric.

All four evaluation criteria for transpiration and for soil moisture were first ranked (1 is best), and then the ranks were summed to select the optimum model parameterization, based on the smallest sum of the ranks. We first calibrated the model for pine and subsequently for cypress. Because of the homogeneous soil physical properties at the study site, we maintained the soil parameter values obtained for the pine tree calibration for the cypress calibration. We used data from December 2020 to August 2021 for calibration and from September 2021 to June 2022 for model validation.

**2.2.5. Comparison of calibrated Noah-MP with default Noah-MP used in WRF**

We conducted a comparison between the water balance components simulated with the calibrated Noah-MP model and those simulated with Noah-MP used within the WRF model, utilizing its default global settings, for both the calibration and validation periods. The default global settings of Noah-MP in WRF for our research site encompass a 2-m soil column, a clay loam soil texture, an open shrubland plant functional type, and specific physics schemes, as listed in Table A1. We refer

to this model parameterization as default Noah-MP. We utilized the initial conditions of pine for running the default parametrized WRF. Furthermore, the open shrublands in the default Noah-MP have a lower *LAI* and *FVEG* than our Noah-MP settings for evergreen needle-leaf trees. The default Noah-MP uses monthly dynamics of *LAI* (ranging from 0.6 in July to 2.58 in January) and *FVEG* (0.17 in July, 0.54 in January), whereas we used constant values for *LAI* (4.0 for pine) and *FVEG* (0.9 for pine) at the research site.

**3. Results**

**3.1. Sensitivity Analysis**

The water balance components for the base scenario of the sensitivity analysis, as a fraction of the precipitation, are presented in Table 2. As it can be seen in the table, the three hydrologic years represent very dry, dry and very wet conditions, as the long-term average precipitation at the study site is around 315 mm. In all three years, transpiration was the

largest water balance component.

The relative changes in the modeled soil evaporation, tree transpiration, and total runoff resulting from changing the input parameters from their base values to their minimum and maximum values are presented in a heat map in Fig. 3. The figure shows that the soil parameters had stronger impacts on the outputs than the vegetation parameters (right side of the heat map is more highlighted). Relative changes were higher for runoff than for the other two outputs, especially in the very wet year.

*RC* values higher than 1 (100%) were all related to runoff (shown in dark red in the map). We can also see that the two dry years behaved very similar.



**Table 2: Simulated annual water balance components for the base parameter values, as a fraction of the precipitation (P); $E_t$ is evaporation from the canopy, $E_g$ is soil evaporation, T is tree transpiration, $R_s$ is surface runoff, $R_d$ is drainage, and ΔSM is soil moisture change.**

| Year | P (mm) | T/P | $E_g$/P | $E_t$/P | $R_s$/P | $R_d$/P | ΔSM/P |
|------|--------|-----|---------|---------|---------|---------|-------|
| 2016-2017 | 49 | 0.76 | 0.31 | 0.14 | 0.01 | 0.00 | -0.21 |
| 2017-2018 | 232 | 0.61 | 0.19 | 0.06 | 0.05 | 0.00 | 0.08 |
| 2018-2019 | 482 | 0.60 | 0.26 | 0.06 | 0.06 | 0.05 | -0.03 |

Except for the drainage parameter (*SLOPE*), which had no impact on any of the three water balance components in the two dry years ( $\left| RC \right|$ < 0.05), all other soil parameters impacted outputs in all three years. *SLOPE* had no impact on transpiration and affected evaporation and runoff only in the very wet year. We found the highest impacts on runoff associated with the soil parameters *SATDK*, *MAXSMC*, *BB* (pore size distribution index) and *REFKDT*. A decrease in the top soil layer depth from 20 cm (base run) to 10 cm increased soil evaporation, on average, by 40% over the three years, whereas an increase from 20 to 30 cm decreased evaporation by 30%. This is most likely due to the higher soil water contents in a 10-cm layer than a 20-cm layer after small rainfall events, resulting in less resistance to evaporation and transpiration. The highest impacts of the vegetation parameters were related to the vegetation fraction (*FVEG*) and minimum stomatal resistance (*RSMIN*). Transpiration was generally less impacted than evaporation by both vegetation and soil parameters.

For the calibration, we selected all parameters that produced relative changes of -0.05 and lower or 0.05 and higher (round off to -0.1 and 0.1 in Fig. 3) on at least one of the output variables, in at least one of the hydrologic years. These were the following eleven parameters: *REFKDT*, *SATDK*, *BB*, *RSMIN*, *FVEG*, *CH2OP*, *SATPSI*, *RGL*, *NROOT*, *TOPT*, and *SLOPE*. The five most impactful parameters on transpiration, in descending order of impact (see Sect. 2.2.3), were *RSMIN*, *FVEG*, *SATDK*, *RGL*, and *SATPSI*. The five parameters with the highest impact on evaporation and runoff combined, also listed in descending order of impact, were *REFKDT*, *SATDK*, *FVEG*, *BB*, and *RSMIN*. Porosity (*MAXSMC*), field capacity (*REFSMC*), and wilting point (*WLTSMC*) were not selected for the calibration, because their values were derived from field observations. For soil thickness scenarios, the base scenario (first soil layer thickness of 20 cm) was selected for the calibration, because it had a zero-water balance error and facilitated direct comparisons with the soil moisture observations, while small positive and negative errors were found in the two other options.





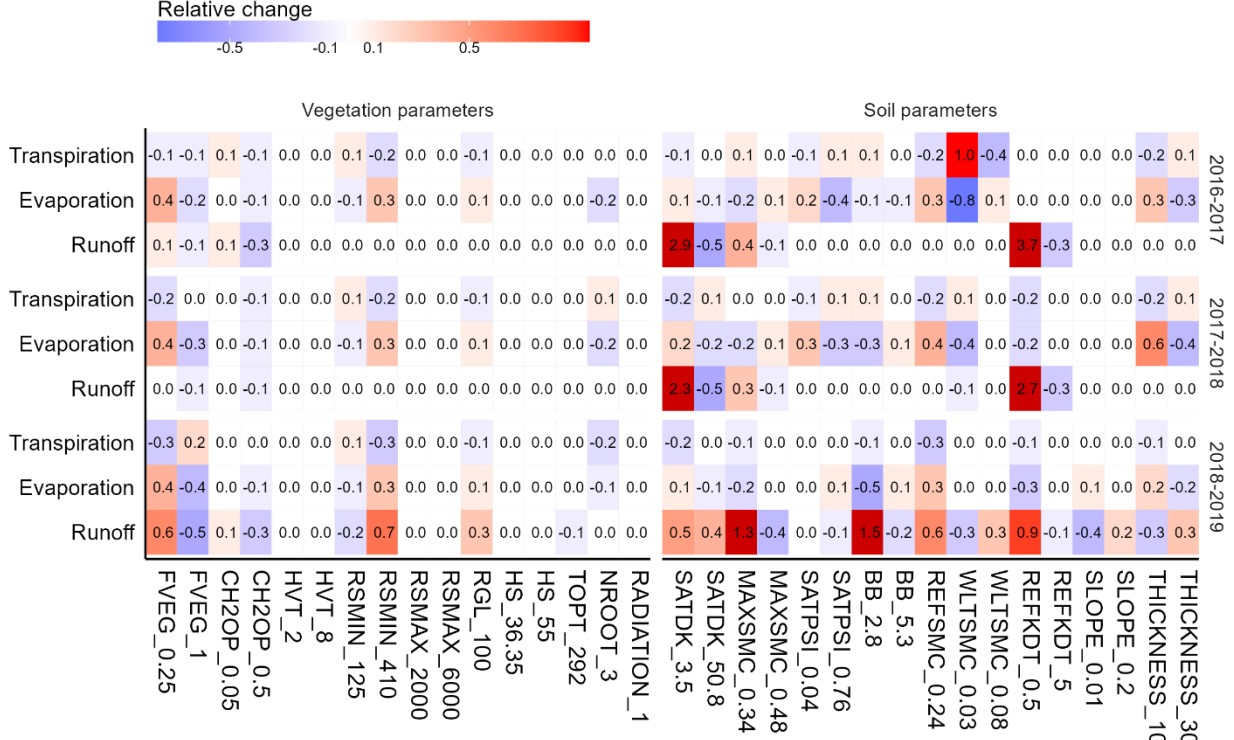

**Figure 3: Relative changes (RC) in tree transpiration, soil evaporation and total runoff (drainage plus surface runoff) resulting from changes in the vegetation and soil parameters for three hydrologic years (2016-2017 to 2018-2019). Absolute relative changes lower than 0.05 (|RC|<0.05) are reported as zero. Relative changes above 1.0 are shown in dark red. RADIATION_1 is the modified two-stream option of the radiation transfer scheme, THICKNESS_10 and THICKNESS_30 runs use a first soil layer thickness of 10 cm and 30 cm (baseline = 20 cm), respectively. For the explanation of the other parameters see Table 1.**

### 3.2. Parameter calibration and model validation

We tested more than 700 different combinations of parameter values and found that the model simulated soil moisture much better than transpiration. This was also the case for the validation period. Table 3 shows the calibrated parameter values for the highest ranked parameterization for pine and cypress. The evaluation criteria for the calibration and validation periods are presented in Table 4.





**Table 3: Calibrated parameters values for the run with the highest ranked evaluation criteria for soil moisture (SM) and for tree transpiration (T) for pine and for cypress.**

| Parameter | Definition | Pine | Cypress |
|---|---|---|---|
| FVEG | Vegetation fraction ($m^2/m^2$) | 0.90 | 0.50 |
| CH2OP | Maximum intercepted water per leaf and stem area (mm) | 0.55 | 0.66 |
| RSMIN | Minimum stomatal resistance (s/m) | 150 | 125 |
| RGL | Radiation stress parameter (-) | 30 | 30 |
| NROOT | Number of soil layers with roots (-) | 4 | 4 |
| TOPT | Optimum transpiration air temperature (K) | 293 | 292 |
| SATDK | Saturated soil hydraulic conductivity (mm/h) | 3.5 | 3.5 |
| SATPSI | Saturated soil matric potential (m) | 0.14 | 014 |
| BEXP | Pore size distribution index (-) | 5 | 5 |
| SLOPE | Drainage parameter | 0.1 | 0.1 |
| REFKDT | Surface infiltration parameter | 0.5 | 0.5 |

**Table 4: Evaluation criteria of the modeled transpiration, average soil moisture (SM) of the 60-cm rootzone, and the soil moisture at 10-, 30-, and 50-cm depth, for pine and cypress, for the calibration and validation periods.**

| Species | Outputs | Calibration | | | | Validation | | | |
|---|---|---|---|---|---|---|---|---|---|
| | | *BIAS\** | *MAE* | *NSE* | *KGE* | *BIAS\** | *MAE* | *NSE* | *KGE* |
| | | *(mm)* | *(mm/d)* | *-* | *-* | *(mm)* | *(mm/d)* | *-* | *-* |
| *Pine* | Transpiration | 44.5 | 0.2 | 13.8 | -1.3 | 17.5 | 0.2 | -0.5 | 0.2 |
| | SM (average) | -0.6 | 1.5 | 0.6 | 0.6 | -1.4 | 2.6 | 0.8 | 0.8 |
| | SM at 10 cm | -6.5 | 6.6 | -0.8 | 0.3 | -8.2 | 8.7 | -0.3 | 0.3 |
| | SM at 30 cm | 1.3 | 1.5 | 0.1 | 0.6 | -0.8 | 3.0 | 0.7 | 0.7 |
| | SM at 50 cm | 3.4 | 3.4 | -22 | -0.2 | 5.2 | 5.2 | 0.2 | 0.5 |
| *Cypress* | Transpiration | 70.5 | 0.4 | -6.7 | -0.7 | 39.2 | 0.2 | 0.0 | 0.5 |
| | SM (average) | -2.6 | 2.8 | -1.7 | 0.2 | -2.8 | 4.0 | 0.7 | 0.8 |
| | SM at 10 cm | -9.0 | 9.0 | -4.8 | -0.3 | -11.0 | 11.0 | -1.4 | 0.1 |
| | SM at 30 cm | -0.3 | 1.8 | 0.2 | 0.6 | 3.9 | 4.3 | 0.6 | 0.7 |
| | SM at 50 cm | 1.5 | 1.6 | -3.5 | 0.0 | 3.5 | 3.7 | 0.5 | 0.6 |

*\*Transpiration BIAS is sum of daily errors (mm) and soil moisture BIAS is average daily errors (mm/d).*

Total precipitation over the 9-month calibration period was 175 mm. For pine, observed transpiration over the 8.1 m$^2$ grid cell area, as derived from the calibrated *FVEG* (0.9), was 62% of the precipitation, which amounts to 109 mm. For cypress, the calibrated *FVEG* was 0.50, which corresponds to a 12.4 m$^2$ grid cell area, and the observed transpiration accounted for

approximately 78% of the precipitation (equivalent to 136 mm) of the precipitation over the grid cell area. However, the fraction of precipitation over the grid cell area that goes to transpiration during the 10-month validation period, with 379 mm of precipitation, was almost the same for pine and cypress (38% for pine and 36% for cypress).

High values of *SATDK* and *REFKDT* result in low runoff and high infiltration and soil moisture, showing overestimation of transpiration and soil moisture during the wet period. This contrasted with the observed transpiration, which showed a much

more even distribution over the season. When both *SATDK* and *REFKDT* were tuned towards their lowest values to fit the observed soil moisture, transpiration was consistently underestimated. Therefore, to model the observed transpiration we needed to set low *RSMIN* values to increase transpiration. Notably, *REFKDT* exhibited particular sensitivity when hydraulic conductivity was very low. Therefore, the best-ranked evaluation criteria for soil moisture and transpiration were achieved with low values for the hydraulic conductivity and infiltration parameter (*SATDK* = 3.9 mm/h, *REFKDT* = 0.5), and *RSMIN*

for pine set to 150. For cypress, we maintained the same soil parameters and *NROOT* values as used for pine. However, the calibrated *RSMIN* in cypress (125) was slightly lower than in pine due to higher transpiration rates of cypress. Additionally, we increased interception (*CH2OP*) to better fit the observed soil moisture during the wet period.

The largest obstacle in simulating transpiration during the calibration period occurred during the period without precipitation (summer time), when soil moisture approached its wilting point, leading to a near absence of modeled transpiration (Fig. 4

and Fig. 5). Despite both observed and modeled soil moisture reaching the wilting point by late April 2021, both species preserved their transpiration levels throughout the summer, with a slight increase in transpiration following the rainfall event in June. The discrepancy between observed and modeled transpiration in both species suggests that the trees extracted water from depths beyond the reach of the soil moisture sensors, possibly below 60 cm. Consequently, this resulted in a considerable positive transpiration bias (44.5 mm in pine and 70.5 mm in cypress) and consistently negative *NSE* and *KGE*

values for transpiration (Table 4).



We attempted to enhance modeled transpiration during the dry period by adjusting the *TOPT* and RGL parameters of the Jarvis scheme. However, modifying *TOPT* had little impact on transpiration, except for a brief increase in the second half of April, which led to poor *NSE* and *KGE* values for transpiration. In contrast, increasing *RGL* from its base value resulted in a slight rise in transpiration from mid-April to mid-June, but also caused a decrease in transpiration during December and

January, ultimately leading to increased underestimation of transpiration and poorer evaluation criteria values for soil moisture. After experimenting with various values, we found that low values of *TOPT* and *RGL* for both species, as shown in Table 3, provided the best results.

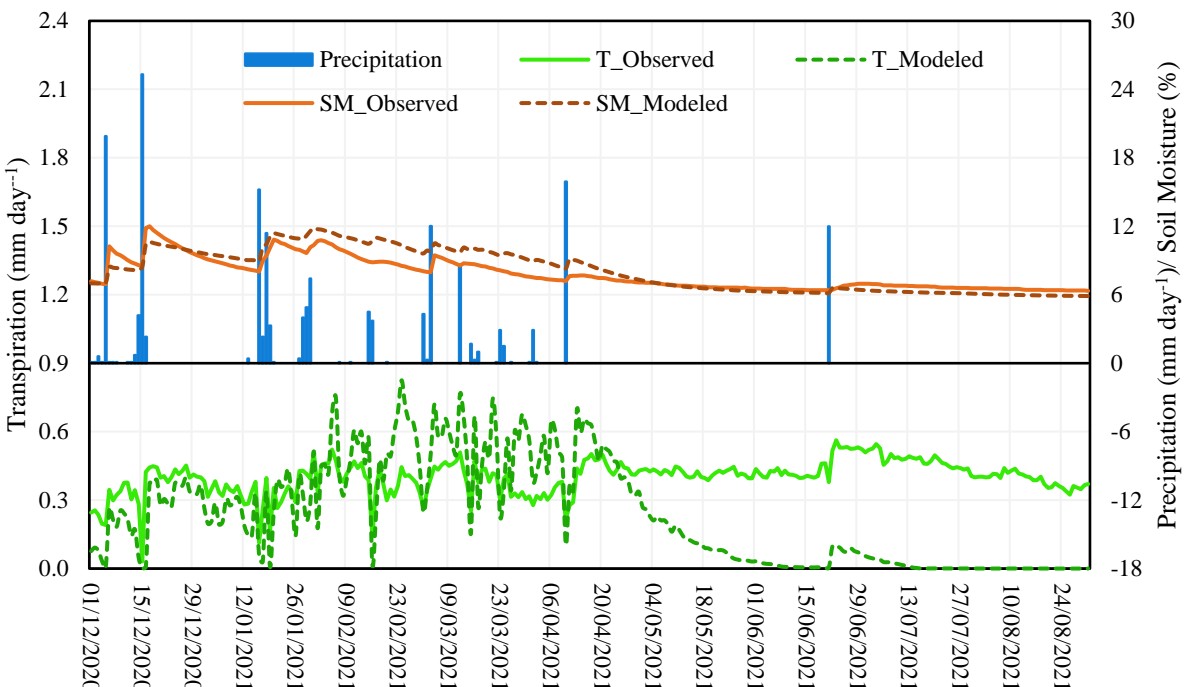

**Figure 4: Daily time series of precipitation and modeled and observed average soil moisture (SM) and tree transpiration (T) for pine, for the calibration period**.





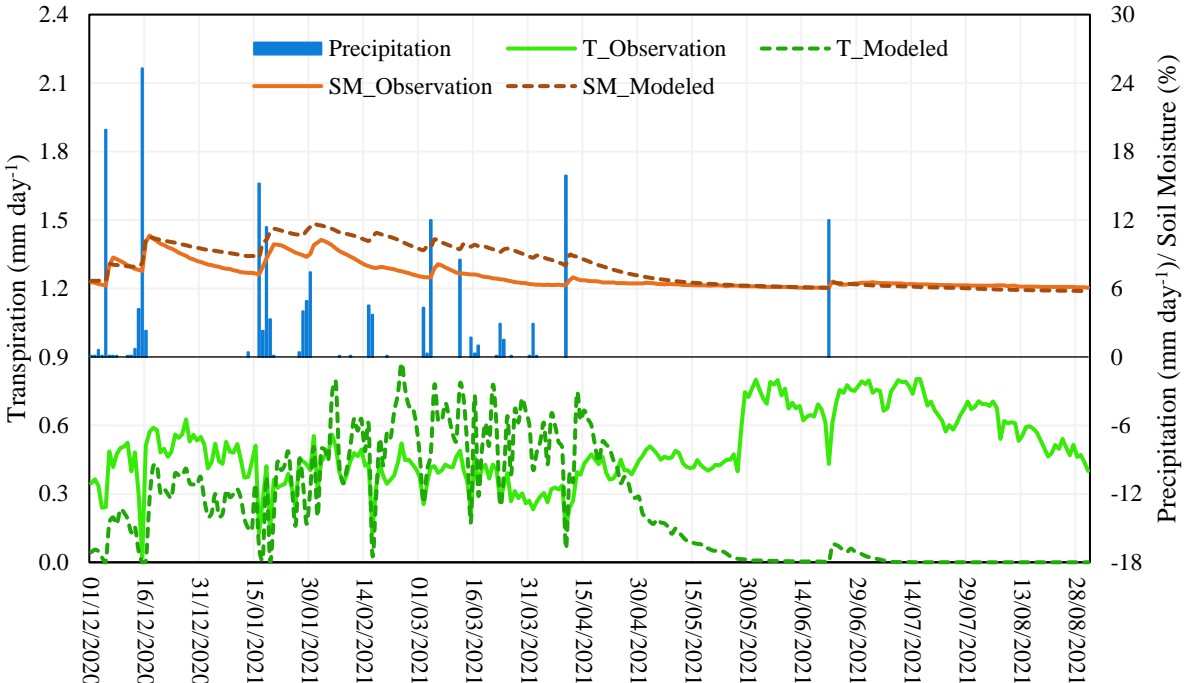

**Figure 5: Daily time series of precipitation and modeled and observed average soil moisture (SM) and tree transpiration (T) for cypress, for the calibration period.**

Figure 6 and 7 illustrate the validation results for pine and cypress. The model demonstrated improved simulation of soil moisture and transpiration during the validation period, primarily due to the presence of more evenly distributed

precipitation compared to the calibration period. Considering that we calibrated the model for the average soil moisture of the three layers, this was, with one exception, always better modeled than the soil moisture in each separate layer, for both species over both periods (Table 4). Among the three layers, the best model performance was observed in the second layer (20-40 cm) for both pine and cypress during both the calibration and validation periods. This is because the model can only use a single set of soil physical properties for the soil column and the daily values of soil moisture in the second layer, unlike

the other two layers, closely aligned with the average daily values of the soil moisture of the three layers. Whereas the model specifies wilting point as the minimum soil moisture content, the soil moisture of the top layer falls below wilting point due to evaporation. This results in a general overestimation of the soil moisture of the top layer, which is subsequently balanced by an underestimation of soil moisture of the third layer (Table 4). The soil moisture of the third layer was better modeled





for cypress, specifically in the validation period (*NSE* = 0.5). Recorded soil moisture in the second and third layers was

lower for cypress than for pine, indicating that cypress has better access to and consumes more water from theses layers.

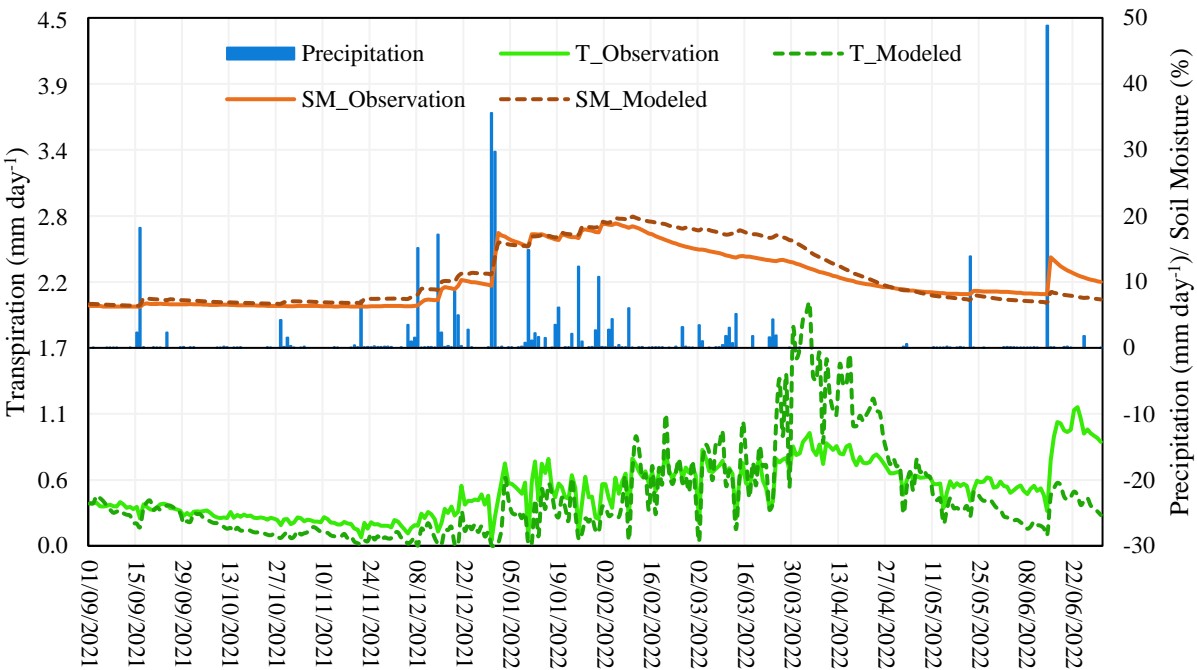

**Figure 6: Daily time series of precipitation and modeled and observed average soil moisture (SM) and tree transpiration (T) for pine for the validation period.**

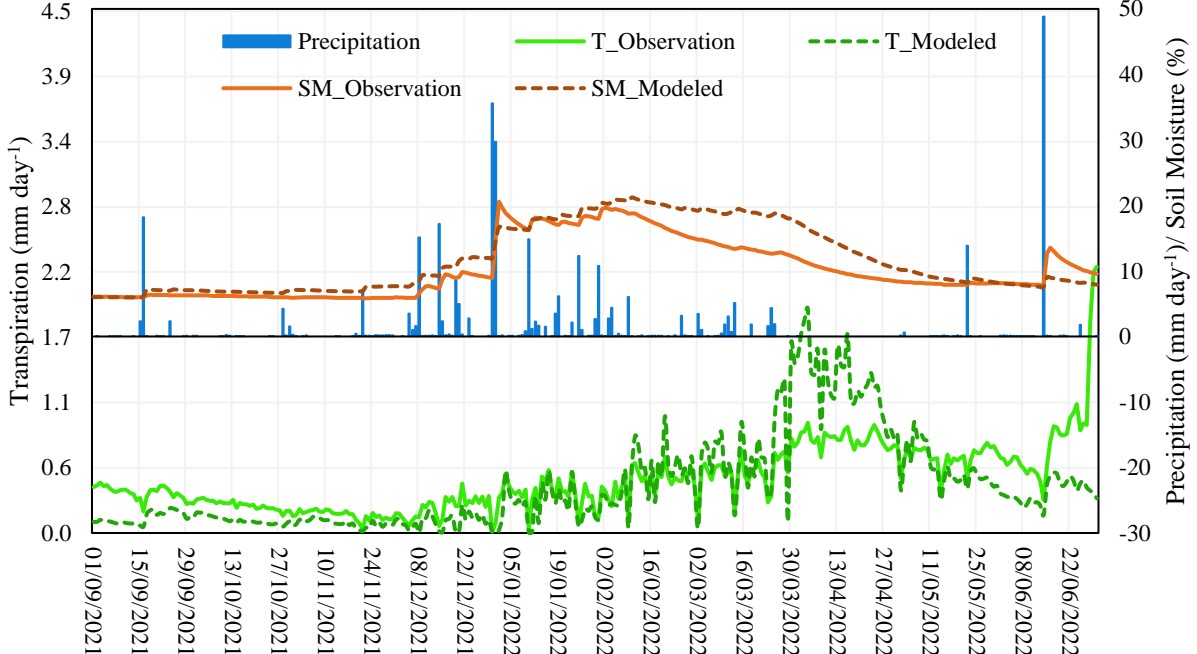

**Figure 7: Daily time series of precipitation and modeled and observed average soil moisture (SM) and tree transpiration (T) for cypress for the validation period.**

Figure 8 shows scatterplots of modeled daily water losses (the sum of modeled daily transpiration, interception, soil evaporation, and runoff) versus observed daily water losses (the observed precipitation minus the soil moisture changes over the day) to better explain the model performance in capturing daily water use in presence and absence of daily rainfall events. The figure shows relatively higher observed than modeled water losses during rainfall events. These losses from the
observed water balance could be related to surface runoff or preferential flows. Because Noah-MP cannot simulate preferential flows, we modeled these losses as surface runoff. Observed negative water losses (water gains), which generally occurred on the day after rain, could also indicate the existence of preferential flows. These preferential flows may have created moist spots at deeper depths that sustained tree transpiration during the dry period. The sum of negative water losses in pine was 8 mm while in cypress it was 17.2 mm, which can support the higher observed transpiration in cypress than pine.





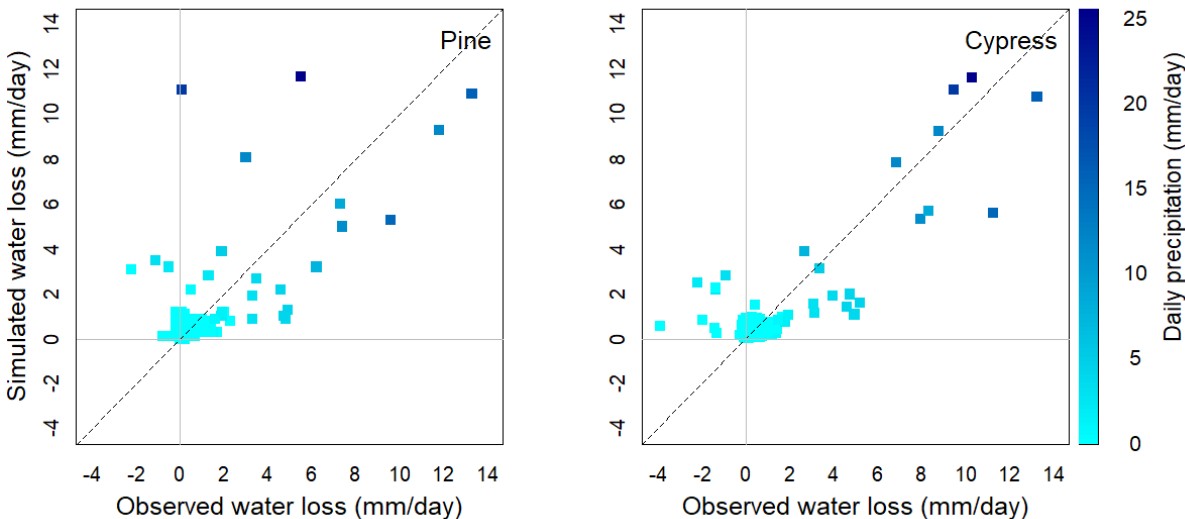

**Figure 8: Observed versus modeled daily water losses in pine and cypress for the calibration period.**

### 3.3. Comparison of calibrated Noah-MP with default Noah-MP used in WRF

Table 5 presents the water balance components simulated with the default Noah-MP, with the calibrated Noah-MP, and those recorded in the plot (observed). The default Noah-MP simulations had much lower transpiration, higher change in soil moisture storage and evaporation, lower interception and lower runoff than the calibrated Noah-MP simulations. The calibrated Noah-MP captured observed soil moisture and transpiration better than the default version, with the latter consistently underestimating transpiration and overestimating soil moisture (Fig. A1 and Fig. A2). Some of these differences arise from the difference in soil texture, with clay loam of the default Noah-MP having a higher water-holding capacity than the sandy loam observed in the field, which consequently resulted in higher soil moisture and less runoff for the default Noah-MP, relative to the calibrated Noah-MP. The simulations show that the modeled soil moisture using the default Noah-MP settings never reached or even came close to the default wilting point of clay loam soil texture (10.3%) even during periods with no precipitation. This is primarily attributed to the very low default LAI values of open shrublands during the summer months (LAI = 0.6 during June-August), which result in a lower rate of transpiration and hence a lower drying rate of the soil. The higher surface runoff simulated with the calibrated Noah-MP may have been trapped in micro-puddles and captured by preferential flows within the grid cell. Total evapotranspiration was higher in all modeling options in the





calibration period (65%-77% of $P$) than in the validation period (53%-58% of $P$). The calibrated model for cypress had the

lowest evapotranspiration due to its lower interception compared to pine and the default model had the highest values of

77% in the calibration period (Table 5).

**Table 5: Water balance components, as a fraction of the precipitation, of pine and cypress during the calibration and validation periods simulated by calibrated Noah-MP, default Noah-MP settings, and recorded in the field. Abbreviations are explained in Table 2.**

| Period | Option | P (mm) | ΔSM/P | T/P | $E_g$/P | $E_t$/P | $R_s$/P | $R_d$/P |
|--------|--------|--------|-------|-----|------|------|------|------|
| Calibration | Noah-MP (*pine*) | 175.4 | -0.04 | 0.38 | 0.09 | 0.25 | 0.33 | 0.00 |
| | Noah-MP (*cypress*) | 175.4 | -0.03 | 0.36 | 0.13 | 0.16 | 0.38 | 0.00 |
| | WRF (default) | 175.4 | 0.17 | 0.26 | 0.47 | 0.04 | 0.05 | 0.00 |
| | Observed (*pine*) | 175.4 | -0.03 | 0.62 | | | | |
| | Observed (*cypress*) | 175.4 | -0.02 | 0.78 | | | | |
| Validation | Noah-MP (*pine*) | 379.4 | 0.00 | 0.32 | 0.05 | 0.21 | 0.42 | 0.00 |
| | Noah-MP (*cypress*) | 379.4 | 0.01 | 0.33 | 0.07 | 0.13 | 0.46 | 0.00 |
| | WRF (default) | 379.4 | 0.38 | 0.22 | 0.28 | 0.04 | 0.08 | 0.00 |
| | Observed (*pine*) | 379.4 | 0.06 | 0.38 | | | | |
| | Observed (*cypress*) | 379.4 | 0.05 | 0.36 | | | | |

## 4. Discussions

The findings of our sensitivity analysis are largely consistent with previous studies, as the same parameters were identified

as sensitive. For example, Arsenault et al. (2018) conducted a global-scale study to identify sensitive parameters in the

Noah-MP model using a global sensitivity method and found the same soil parameters we identified as having high impact

on evapotranspiration and runoff. They examined both dynamic and prescribed leaf area index (*LAI*) vegetation options in

their study and found *RSMIN*, *RGL* and *TOPT* to be sensitive parameters. However, in contrast to our findings, they also

identified *HS* (vapor pressure deficit parameter) as sensitive. This difference is most likely due to the strong water limitation

in our study area. Cuntz et al. (2016) found in their study on the sensitivity of the Noah-MP parameters over 12 catchments

in the US that surface runoff was very sensitive to *REFKDT* and *SATDK*. Similarly, we found high relative changes in runoff

for these two parameters. They also identified *BB*, *MAXSMC*, and *REFSMC* as sensitive model parameters for all outputs.

However, in contrast to our findings, both previous studies found *CH2OP* and *SATPSI* to be insensitive, although our

relative changes for *SATPSI* exceeded 0.1 only in the two dry years. Sofokleous et al. (2023) reported *SATDK* and *MAXSMC* as sensitive parameters for both simulated runoff and evapotranspiration in 31 mountainous watersheds of Cyprus using Noah-MP. These authors also found *SLOPE* to affect these two variables, which was not seen here because *SLOPE i*s related to the drainage at the bottom of the soil column, which is not often observed in the harsher semi-arid conditions of the plains.

The calibration resulted in a *CH2OP* to 0.55 mm (per unit *LAI*) in pine, which gave an interception rate of 25% and 21% of the precipitation in the calibration and validation periods, respectively. These modeled interception rates for pine align with the observed and modeled interception rates reported by Eliades et al. (2022). The authors reported observed interception rates ranging from 13% to 55% of the precipitation over 12 years (2008-2019) for a stand in the foothills of Cyprus' Troodos Mountains, with an average rainfall of 429 mm. Higher interception was associated with drier years, with the highest interception (55%) occurring in the driest year (186 mm) and the lowest interception (13%) in a wet year (475 mm). Similarly, we found higher interception (25%) during the drier calibration period and lower interception (21%) during the wetter validation period.

We found an underestimation in evapotranspiration in the model, which is also discussed in previous studies conducted in arid and semi-arid regions. For example, Ma (2023) applied the Noah-MP model to estimate water and energy fluxes in two representative alpine meadow and steppe ecosystems on the Tibetan plateau and found an underestimation of evapotranspiration for both ecosystems. The author incorporated a nonlinear (asymptotic) root distribution function in the model, which improved the evapotranspiration estimates and the partitioning between transpiration and soil evaporation in the alpine meadow, increasing the daily *NSE* for evapotranspiration from 0.84 to 0.90. However, the improvement in the alpine steppe was marginal (from -0.45 to -0.37). The author related the poor evapotranspiration simulations in the alpine steppe to a sparse vegetation cover.

Zheng et al. (2015) modeled soil moisture using Noah-MP in three different soil depths of 5, 25, and 70 cm in a Tibetan meadow ecosystem. In line with our findings (Table 4), they found a consistent overestimation of soil moisture in the top soil layer and an underestimation in the lowest layer. Similar to Ma (2023), Zheng et al. (2015) incorporated two nonlinear root distribution functions (exponential and asymptotic) in the model. Using either of the nonlinear equations, they were able to



improve water uptake from different soil layers. Such functions could possibly also improve our simulations, considering that cypress relies heavily on lateral roots concentrated in the subsurface, as Rog et al. (2021) reported. These authors found that root growth of *C. sempervirens* is mostly horizontal and that of *P. halepensis* is both horizontal and vertical, based on

root sampling in top 20 cm soil on limestone bedrock and allometric equations.

## 4. Conclusions

This study used the Noah-MP model to simulate the water balance components of two conifer species, *Pinus brutia* and *Cupressus sempervirens*, in an eastern Mediterranean ecosystem. The model's performance was also compared to those simulated with the default Noah-MP settings in the WRF model for the research site. Based on a local sensitivity analysis,

vegetation fraction (*FVEG*), minimum stomatal resistance (*RSMIN*), surface infiltration parameter (*REFKDT*), and saturated soil hydraulic conductivity (*SATDK*) were identified as the parameters with the highest impacts on the transpiration and soil water balance components that were subsequently calibrated in the model.

Noah-MP showed improved performance during the wetter validation period (379 mm rain) compared to the drier calibration period (175 mm rain) for both species, effectively capturing the average soil moisture, as observed with 24 soil moisture

sensors for each species. The middle soil layer exhibited better modeling performance compared to the other layers. Soil moisture in the first and third layers was over- and underestimated, respectively, which can be attributed to the model's use of a single set of soil physical properties for the soil column and a uniform root distribution for the rootzone. The single set of soil properties also fixes the lowest soil moisture content at wilting point, while in reality the top soil layer becomes near air dry. The model failed to capture the observed tree transpiration, although positive *KGEs* (0.2 for pine, 0.5 for cypress)

were obtained during the validation period. This is most likely due to heterogeneous wetting and water uptake in the soil profile.

Comparison between the calibrated and the default Noah-MP models revealed that runoff in the calibrated model was significantly higher than in the default model, which can be attributed to the higher water holding capacity of clay loam in the default Noah-MP, compared to that of the sandy loam soil of the research site, as used in the calibrated model. For both

model parameterizations, the evapotranspiration, as a fraction of the precipitation, was higher during the calibration period



than during the validation period, with the highest values associated with the default Noah-MP and the lowest values associated with the cypress in the calibrated model. Incorporating a nonlinear root distribution function could potentially improve model performance by providing a more accurate estimation of plant water use from different soil layers, resulting in better soil moisture estimation and improved estimation and partitioning of evapotranspiration. Additionally, installing soil moisture sensors below the 60-cm depth could help improve our understanding of the vertical root extension and water extraction for transpiration.



## Appendix A

**Table A1: Noah-MP multi-physics schemes with their possible options, the options used for the sensitivity analysis, calibration and validation (Selected) and the options of the default Noah-MP, as used in WRF.**

| Schemes | Description | Options | Selected options | Default options |
|---|---|---|---|---|
| **OPT_DVEG** | Vegetation model | 1. Prescribed [table LAI, shdfac=FVEG] | 1 | 1 |
| | | 2. dynamic together with OPT_CRS = 1 | | |
| | | 3. table LAI, calculate FVEG | | |
| | | 4. table LAI, shdfac=maximum | | |
| **OPT_CRS** | Canopy stomatal resistance | 1. Ball-Berry | 2 | 2 |
| | | 2. Jarvis | | |
| **OPT_BTR** | Soil moisture factor for stomatal Resistance | 1. Noah soil moisture | 1 | 1 |
| | | 2. CLM matric potential | | |
| | | 3. SSiBb matric potential | | |
| **OPT_RUN** | Runoff and groundwater | 1. SIMGM, TOPMODEL with groundwater | 3 | 3 |
| | | 2. SIMTOP, TOPMODEL with an Equilibrium water table | | |
| | | 3. Schaake96, original surface and subsurface runoff (free drainage) | | |
| | | 4. BATS, surface and subsurface runoff (free drainage) | | |
| **OPT_SFC** | Surface layer drag coeff (CH & CM) | 1. M-O | 1 | 2 |
| | | 2. Chen97, Original Noah | | |
| **OPT_FRZ** | Super cooled liquid water | 1. NY06, no interaction | 1 | 1 |
| | | 2. Koren99, Koren's interaction | | |
| **OPT_INF** | Frozen soil permeability | 1. NY06, linear effect, more permeability | 1 | 1 |
| | | 2. Koren99, nonlinear, less permeability | | |
| **OPT_RAD** | Radiation transfer | 1. Modified two-stream, Gap < 1- FVEG | 1 and 3 | 1 |
| | | 2. Two-stream, Gap = 0 | | |
| | | 3. Two-stream, Gap = 1- FVEG | | |
| **OPT_ALB** | snow surface albedo | 1. BATS | 2 | 2 |
| | | 2. CLASS | | |
| **OPT_SNF** | rainfall & snowfall | 1. Jordan91 | 1 | 3 |
| | | 2. BATS, SFCTMP<TFRZ+2.2 | | |
| | | 3. Noah, SFCTMP<TFRZ | | |
| **OPT_TBOT** | lower boundary of soil temperature | 1. zero-flux | 2 | 2 |
| | | 2. Noah | | |
| **OPT_STC** | snow/soil temperature time scheme (only layer 1) | 1. semi-implicit | 1 | 1 |
| | | 2. fully implicit, original Noah | | |



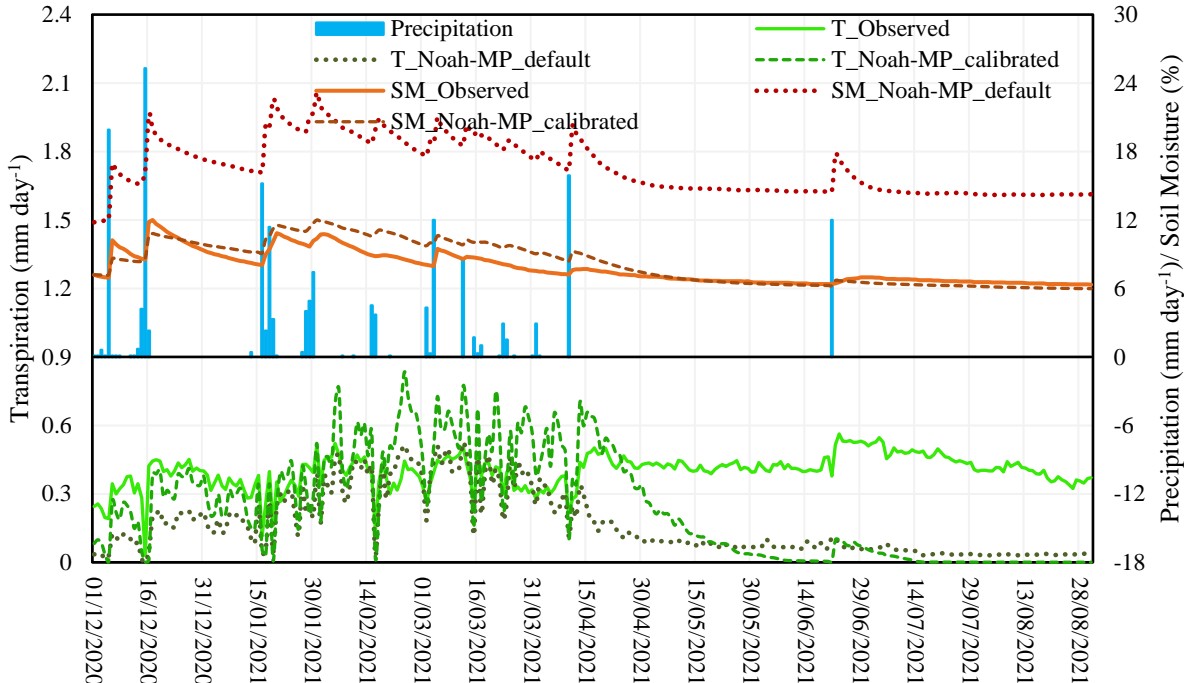

**Figure A1: Figure A1: Daily time series of observed precipitation, average soil moisture (SM) and tree transpiration (T) and the**
**400**  **modeled SM and T for pine with the calibrated Noah-MP and for open shrubland with the Noah-MP default, for the calibration**
**period. The initial soil moisture content of the clay loam soil of the default Noah-MP was set equal to its wilting point (10.3 %) plus**
**the initial moisture content of the sandy loam soil of the calibrated Noah-MP in the calibration period (7.2 %) minus its wilting**
**point (6.0 %), resulting in an initial moisture content of (11.5 %).**



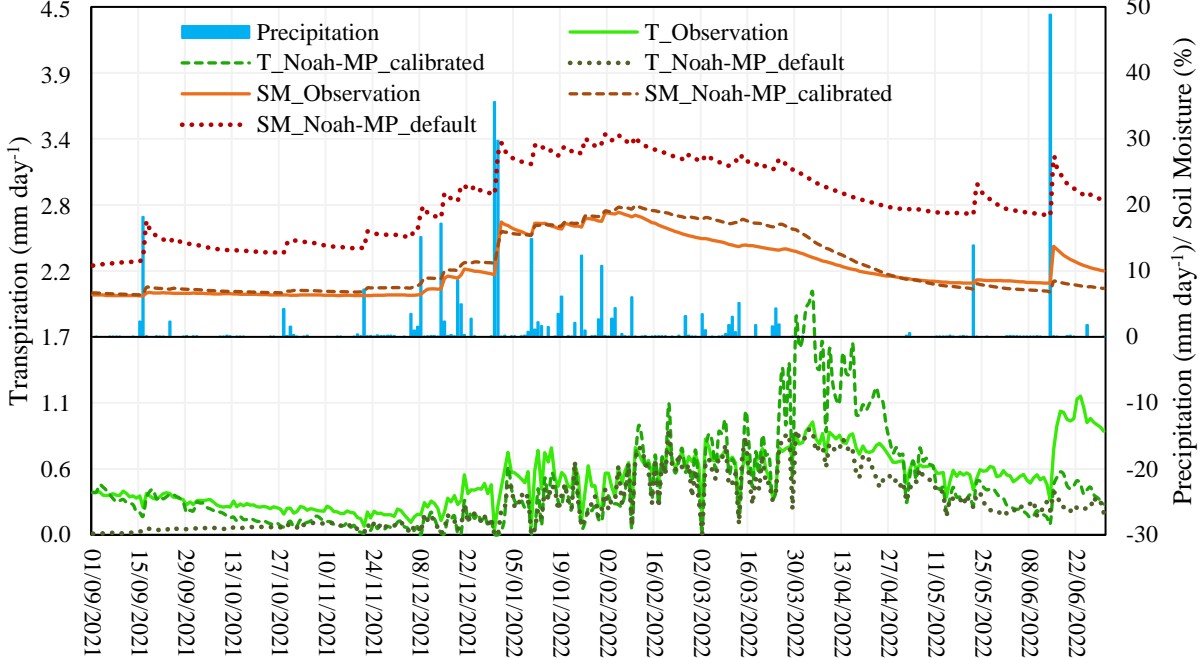

**Figure A2: Daily time series of observed precipitation, average soil moisture (SM) and tree transpiration (T) and the modeled SM and T for pine with the calibrated Noah-MP and for open shrubland with the Noah-MP default, for the validation period. The initial soil moisture content of the sandy loam soil was 6.9 % and that of the clay loam soil was set at 11.2 %, as described in Figure A1.**

**Code availability**

The codes used in the development of all analyses will be made available upon request. The code used to run the Noah-MP

land surface model can be found at **https://ral.ucar.edu/model/noah-multiparameterization-land-surface-model-noah-**

**mp-lsm**

**Data availability**

The model simulation data from this study are available on: **https://doi.org/10.5281/zenodo.10900317**.

Field data will be made available on request.

**Author contribution**

**Mohsen Amini Fasakhodi:** Conceptualization, Data curation, Formal analysis, Investigation, Methodology, Software,
Validation, Writing-Original Draft. **Hakan Djuma:** Data Curation, Formal Analysis, Investigation, Methodology, Writing-Review and Editing. **Ioannis Sofokleous:** Data Curation, Formal Analysis, Investigation, Software, Writing-Review and Editing. **Marinos Eliades:** Investigation. **Adriana Bruggeman:** Conceptualization, Funding acquisition, Methodology, Supervision, Writing-Review and Editing.

**Competing interests**

The authors declare that they have no competing interests that could have influenced this work.

**Acknowledgments**

We would like to express our sincere thanks to the Cyprus Department of Forests and to the Cyprus Department of Meteorology for their support of this research. This research has received financial support from the PRIMA (2018 Call) SWATCH Project and the Water JPI (Joint Call 2018) FLUXMED Project, both funded by the Republic of Cyprus through
the Cyprus Research and Innovation Foundation. The PRIMA programme is supported by Horizon 2020, the European Union's Framework Program for Research and Innovation.

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
