# Peer review of "Modeling water balance components of conifer species using the Noah-MP model in an eastern Mediterranean"

_Hydrology and Earth System Sciences, 2024_

## Author Response (AR2)

**Authors Responses**

Dear Editor and Reviewers,

We sincerely thank you for your time and effort in reviewing our manuscript. Your comments and suggestions have been invaluable in improving the quality of our work. Below, we provide a point-by-point response to each comment.

**Reviewer 1**

**Comment 1:** My main concern is about the number of sap flow sensors, only two sap flow sensors per species, which in my opinion is quite small. To overcome this significant limitation, you must make clear that transpiration variability within species in the plantation is sufficiently low. Additionally, concerning tree variability, the standard error for tree dimensions is missing. Providing this information will inform readers about the representativeness of the studied trees within the plantation.

**Response:** To address this concern, we have added a new paragraph (lines 121-129) to our manuscript which provides additional information on tree selection.

**Response in the manuscript:**

Sap flow was monitored on six pine and six cypress trees at the site. The mean and standard deviation of the stem diameter of the monitored trees were 9.2 ± 1.2 cm for pine, and 10.3 ± 1.7 cm for cypress, which were close to the full field averages of 10.3 ± 2.3 cm for pine, and 9.9 ± 1.8 cm for cypress (Djuma et al., under review). Soil moisture was monitored in the rootzone of two pine trees and two cypress trees with sap flow sensors. These four trees are used for the current study. The mean stem diameter of the 4 studied trees (9.5 cm pine, 9.3 cm cypress) was similar to the 12-tree averages. The mean total sap flow of the 4 trees (296 mm pine, 566 mm cypress) during the December 2020 to June 2022 study period was also reasonably close to the 12-tree averages (314 mm pine, 642 mm cypress). The closer fit of the pine tree means was indicative of the lower sap flow variability of these trees (215 mm-357 mm), compared to cypress (405 mm-1061 mm) (Djuma et al., under review). These numbers suggest that the 4 trees used in this modeling study were representative of the trees at the site.

**Comment 2:** It would be important to have some more information concerning the water use strategies and rooting systems of the studied species, *Pinus brutia* and *Cupressus sempervirens*, in the introduction (eventually around lines 40-48), to help results interpretation and subsequent discussion.

**Response:** To address this issue, we have added a new paragraph (lines 49-61) to our manuscript, where we add findings from studies on provide a brief overview of the differences in rooting systems between the two species:

**Response in the manuscript:**

Although *P. brutia* and C. sempervirens have functional similarities in Mediterranean environments, their root systems differ significantly. Research by Rog et al. (2021) in a hot Mediterranean climate (510 mm annual rainfall) revealed that *P. halepensis* (similarities to *P. brutia* mentioned above) develops deep, extensive roots, accessing water from lower soil layers. In contrast, *C. sempervirens* has a shallower root system, potentially relying more on surface moisture. This study reported that *P. halepensis* roots were found throughout the shallow terra rossa soil (average

depth 21 cm) and cracked limestone bedrock, while *C. Sempervirens* had fewer roots in bedrock. These findings about *P. halepensis* align with Eliades et al.'s (2018) observations about *P. brutia*. They studied a homogeneous *P. brutia* forest in Cyprus's Troodos mountains, characterized by shallow soil, fractured bedrock, and 425 mm average annual rainfall. They discovered that these trees' root systems extend horizontally up to 10 meters and frequently grow within bedrock fractures. A study by Ares and Peinemann (1992) examined fine roots' quantity and vertical distribution in 12 coniferous plots, including four stands of *P. halepensis* and two stands of *C. sempervirens*. The study area has a temperate climate and loess soil with a depth ranging from a few centimeters to 120 cm overlying a bedrock. Their findings showed that in the upper 50 cm of the soil profile, *C. sempervirens* had higher fine-root biomass than *P. halepensis* in similar forest-quality classes.

**Comment 3:** The mention of the reported or studied water balance components (transpiration and soil moisture?) in the abstract is advised.

**Response:** To address this issue, we have revised the abstract to explicitly mention the specific water balance components that were investigated, namely soil moisture and tree transpiration.

**Response in the manuscript:**

To this end, we  calibrated the model for rootzone soil moisture and transpiration of two conifer species, *Pinus brutia,* and *Cupressus sempervirens,* in a plantation forest on the Mediterranean island of Cyprus.

**Comment 4:** The choice of the parameter designation is not very straightforward (e.g. REFKDT or SATDK), the manuscript would benefit with the use of simpler designations or the name of the parameters.

**Response:** We appreciate the reviewer's feedback regarding the clarity of parameter designations. We understand that using abbreviations common in model documentation might pose challenges for readers unfamiliar with NOAHMP and WRF. However, these parameter short names (abbreviations) are used in the model documentation and code. They are also used in other journal articles on the NOAHMP and WRF models. All parameters are also listed with their full description and units in Table 1, so these could be relatively easy to find for the reader. For the surface infiltration parameter REFKDT, which is a scaling factor, we added the equation and description in a footnote of Table 1.

**Response in the manuscript:**

**REFKDT is the reference soil conductivity scaling factor which is computed using the following equation: REFKDT = (KDT× REFDK)/ SATDK, where KDT and REFDK are the scaled and reference values for SATDK, respectively.

**Comment 5:** Figure 1: it would be interesting to distinguish between pine and cypress at the site, and eventually identify which trees were used for sap flow

**Response:** We added a few lines (117-120) in the paragraph just above Figure 1 to present the number of all trees, and the number of all pines and cypresses. We also improved Figure 1.

**Response in the manuscript:**

The field is relatively flat (with a mean slope of 4%) and is covered by a combination of seasonal vegetation and, more than 2300 mixed indigenous trees and shrubs with a 5 to 6-m planting distance, which covers a 7.5-ha area. The study site comprises 846 *P. brutia* and 216 *C. sempervirens* trees, allocated to the planting holes without any specific pattern. We will refer to these species as pine and cypress.

[Figure]

**Figure 1: The eastern Mediterranean with the island of Cyprus (red framed) and the study area location identified with a red  bullet (a); the study area in Athalassa National Forest Park framed with a white polygon and the two monitoring locations (each with a pine and cypress tree) identified with red pins (b); and two of the monitored tree species with sap flow and soil moisture sensors (c). © Google Earth.**

**Reviewer 2**

**Comment 1:** The Introduction does not adequately discuss the potential implications of the study's findings. There is no mention of how the results might contribute to model development or forest management practices. Also, the introduction mentions several studies but does not critically engage with them to highlight what has been done and what remains to be explored. It is not clear how the research gap (I assume it is "no research in the literature has combined field observations with land surface models (LSMs) to estimate water balance components in Mediterranean coniferous forests.") would be relevant for new understanding or methodological advances.

**Response:** To address this issue, we made several revisions to the introduction section. We included new information in the third paragraph. Additionally, we elaborated on the potential applications of our findings, highlighting how they can inform the development of improved models and enhance forest management practices.

**Response in the manuscript:**

Despite numerous water balance-related studies in Mediterranean forest areas (e.g., Molina et al., 2012; Montaldo et al., 2021; Simpson et al., 2022), only a few studies have focused on quantifying water balance components of *C. sempervirens, P. brutia* and *P. halepensis* species (e.g., Yaseef et al., 2010; Ungar et al., 2013; Klein et al., 2014; del Campo et al., 2014; Helman et al., 2017b; Eliades et al., 2018; Rohatyn et al., 2018). While some studies have combined observations with modeling techniques, none have specifically examined species-specific water balance components, considering the complexities of soil, vegetation, and atmospheric dynamics. For example, Klein et al. (2014) conducted a study in the 40-year-old Yatir *P. halepensis* forest in Israel, which has a mean annual rainfall of 285 mm and a light brown Rendzina soil overlying a limestone bedrock. They estimated the transpirable soil water content using sap flow, water potential, and depth-dependent water retention curves. They also analyzed the impact of soil layer-specific values of soil parameters on soil water dynamics with the ecosystem gas exchange model MuSiCA. However, they did not calibrate above-ground species-specific vegetation parameters but instead relied on previous studies. In another study conducted in the same forest, Helman et al. (2017b) combined a remote sensing-based model (RS–Met) with eddy covariance observations to estimate the annual evapotranspiration of the planted pine forest.  However, they only used a water deficit factor to adjust their model for natural water-limited ecosystems, ~~Eliades et al. (2018) used sap flow measurements from eight trees, soil moisture, and throughfall observations, and a water balance model to calculate the daily water balance components of a stand of homogenous pine forest (*P. brutia*) in the Troodos mountains of Cyprus under 425 mm average annual rainfall. Rohatyn et al. (2018) used a mobile lab (with flux measurements and meteorological sensors) and a permanent flux~~

. Nonetheless,

no research in the literature has combined field observations with land surface models (LSMs) to estimate water

balance components in Mediterranean coniferous forests. Such studies can improve modeling applications in these

environments, supporting foresters with information on the water use of different species under different climate

conditions and thereby optimizing species selection and planting densities.

**Comment 2:** In the Discussion and Conclusions, while the comparison with previous studies is informative, it lacks depth in explaining why certain differences or similarities occur. For instance, it is still not clear why strong water limitation in your study area might influence these sensitivities differently than in other regions. Still, the broader implications of the sensitivity analysis findings for model calibration, validation, and application are not sufficiently discussed. It would be appreciated to discuss how your results contribute to the understanding of the effects of soil and vegetation parameters on water balance components under different climatic conditions.

**Response:** To address this issue, we made several revisions to the discussion and conclusion sections. In the discussion section, we added a few lines to the first paragraph (lines: 377-387), inserted a new paragraph between the second and third paragraphs (lines: 396-404), and added a few lines at the beginning and to the end of the third paragraph (lines: 405-415). Additionally, in the conclusion section, we expanded the first paragraph (lines: 442-451) to incorporate these changes.

**Response in the manuscript:**

Lines: 377-387

. The sensitivity analysis of our research revealed that *RSMIN*, *REFKDT*, and *SATDK*, had the highest impact on evapotranspiration and runoff, which is consistent with previous studies (e.g., Arsenault et al., 2018; Cuntz et al., 2016). However, we also found that the strong water limitation in our study area influences the sensitivities of parameters differently than in other regions. For instance, Arsenault et al. (2018) conducted a global-scale study to identify sensitive parameters in the Noah-MP model using a global sensitivity method and found the same soil parameters we identified as having a high impact on evapotranspiration and runoff. They examined dynamic and prescribed leaf area index (*LAI)* vegetation options in their study and found *RSMIN*, *RGL,* and *TOPT* to be sensitive parameters. However,  while they identified *HS* (vapor pressure deficit parameter) as a sensitive parameter, we did not find it significant in our study. This discrepancy is likely due to the unique hydro-climatic conditions of our study area, which is characterized by strong water limitation and implies that trees may be more limited by soil water availability than by atmospheric conditions in such ecosystems.

Lines: 396-404

145   The findings of our study, in conjunction with those of other research conducted on a global or national scale, highlight the importance of certain model parameters, regardless of the hydro-climatic conditions. Specifically, the consistency in identifying *REFKDT, SATDK*, and *MAXSMC* as highly sensitive soil parameters across different studies underscores their crucial role in controlling water infiltration and consequently influencing soil water balance. Similarly, *RSMIN*, a vegetation parameter that regulates tree water consumption, was consistently identified as a

150   sensitive parameter in our study and the studies mentioned above-. However, our study also highlights the importance of ecosystem-specific conditions in determining the sensitivity of certain parameters, such as *CH2OP*, which was identified as sensitive in our study but not in others, The varying sensitive parameters identified in different studies suggest that the sensitivity of model parameters and the magnitude of their sensitivity can be highly dependent on the specific characteristics of the study area.

155   Lines: 405-415

Ecosystem-specific conditions, such as soil and vegetation characteristics, imply the need for a unique calibration of identified sensitive parameters and, model parameterization that accurately reflects the reality of the ecosystem. For instance, our calibration resulted in a *CH2OP* of 0.55 mm (per unit *LAI*) in pine, which gave an interception rate of 25% and 21% of the precipitation in the calibration and validation periods, respectively. These modeled interception

160   rates for pine align with the observed and modeled interception rates reported by Eliades et al. (2022). The authors reported observed interception rates ranging from 13% to 55% of the precipitation over 12 years (2008-2019) for a stand in the foothills of Cyprus' Troodos Mountains, with an average rainfall of 429 mm. Higher interception was associated with drier years, with the highest interception (55%) occurring in the driest year (186 mm) and the lowest interception (13%) in a wet year (475 mm). Similarly, we found higher interception (25%) during the drier calibration

165   period and lower interception (21%) during the wetter validation period. The agreement with the findings of Eliades et al. (2022), who conducted their study in a nearby *Pinus brutia* forest, suggests that the calibrated value of *CH2OP* in Noah-MP can be applied to similar conditions in future studies.

lines: 442-451

This study used the Noah-MP model to investigate the water balance components of two conifer species, Pinus brutia

170   and Cupressus sempervirens, in an eastern Mediterranean ecosystem. The model's performance was also compared to

those simulated with the default Noah-MP settings in the WRF model for the research site.  Our findings highlight the importance of sensitive parameters in water balance simulations, with vegetation fraction (FVEG), minimum stomatal resistance (RSMIN), surface infiltration parameter (REFKDT), and saturated soil hydraulic conductivity (SATDK)  having the most significant impacts on transpiration and soil water balance components.  Our sensitivity analysis and subsequent calibration of these parameters demonstrate the potential to improve the accuracy of water balance predictions in similar ecosystems, ultimately contributing to a better understanding of the impact of sensitive parameters on water balance components and informing the development of forest management strategies.

**Comment 3:** In addition, the study observes better model performance during the wetter validation period compared to the drier calibration period. I suggest that an additional calibration-validation framework can be considered. Specifically, the authors could calibrate the model using data from the wetter period (September 2021 to June 2022) and validate it using data from the drier period (December 2020 to August 2021). By reversing the calibration and validation periods, the study can assess whether the model is robust across different hydrological conditions.

**Response:** We reversed the calibration-validation periods and ran the model with the calibration settings over the validation period to evaluate its performance, and tried to retune the model to get better criteria. In the data and model section, we added a sentence to lines 246-248 to provide additional context, presented the results in lines 343-349, and added a new paragraph in the discussion section (lines: 416-423). Additionally, we identified an error in Table 4 and corrected it in the revised version, ensuring the accuracy of the data presented.

**Response in the manuscript:**

Lines 246-248:

Furthermore, we also reversed the calibration and validation periods, using data from September 2021 to June 2022 for calibration and from December 2020 to August 2021 for model validation.

Lines 343-349:

Results for the reversed calibration-validation test showed small improvements for FVEG=0.95 for pine, compared to the original calibrated value of 0.90, in five of the evaluation criteria: soil KGE, BIAS, and MEA improved to 0.9, -1.1 mm/day, and 2.5 mm/day, respectively, and transpiration NSE and KGE increased to -0.4 and 0.3. We also observed improvements for cypress, for FVEG=0.65 instead of the original value of 0.50, in two evaluation criteria of transpiration NSE and KGE (0.5 and -0.1) while the soil evaluation criteria remained nearly equal. However, when

we validated the model using the drier period from December 2020 to August 2021, six evaluation criteria decreased for pine, and seven criteria decreased for cypress, relative to the original calibrations.

lines: 416-423

205  The model's overall better performance when calibrated on the dry year and validated on the wet year, instead of the other way around, suggests that the model captures the relationship between soil moisture and tree transpiration better during drier periods than during wetter periods. The better relationship between evapotranspiration and soil moisture in drier soils than in wetter soils in land surface models is also mentioned in other studies. Larsen et al. (2016) calibrated the SWET land surface model using eddy covariance fluxes and catchment runoff over three different

210  surface types (forest, grass, and agriculture) in Denmark. They found a less distinct relationship between evapotranspiration and rootzone soil moisture in grassland with higher soil moisture compared to the two other surfaces with lower soil moisture.

**Comment 4:** Last but not least, the study only uses a few site observations, which raises my concerns about the generalizability of the findings. While this may not be the focus of the study, modeling only a

215  single tree may lead to an oversimplified representation of the ecosystem. I recommend discussing the limitations of this approach and suggesting ways to improve the generalizability of future studies.

**Response:** A similar point was also raised by Reviewer 1 (Comment 1). To address this issue and to demonstrate the representativeness of the selected trees, we added a paragraph to the manuscript (lines 121-129).

220  **Response in the manuscript:**

Sap flow was monitored on six pine and six cypress trees at the site. The mean and standard deviation of the stem diameter of the monitored trees were 9.2 ± 1.2 cm for pine, and 10.3 ± 1.7 cm for cypress, which were close to the full field averages of 10.3 ± 2.3 cm for pine, and 9.9 ± 1.8 cm for cypress (Djuma et al., under review). Soil moisture was monitored in the rootzone of two pine trees and two cypress trees with sap flow sensors. These four trees are used for

225  the current study. The mean stem diameter of the 4 studied trees (9.5 cm pine, 9.3 cm cypress) was similar to the 12-tree averages. The mean total sap flow of the 4 trees (296 mm pine, 566 mm cypress) during the December 2020 to June 2022 study period was also reasonably close to the 12-tree averages (314 mm pine, 642 mm cypress). The closer fit of the pine tree means was indicative of the lower sap flow variability of these trees (215 mm-357 mm), compared to cypress (405 mm-1061 mm) (Djuma et al., under review). These numbers suggest that the 4 trees used in this

230  modeling study were representative of the trees at the site.